# Laboratory performance prediction using virtual reality behaviometrics

**Philip Wismer**[1☯]**, Sarah Aparecida Soares**[1☯]**, Kasper Alnor Einarson**[2☯]**, Morten Otto Alexander Sommer**[1]*

1 Novo Nordisk Foundation Center for Biosustainability, Technical University of Denmark, Kgs, Lyngby, Denmark, 2 Department of Applied Mathematics and Computer Science, Technical University of Denmark, Kgs, Lyngby, Denmark

☯ These authors contributed equally to this work.
* msom@bio.dtu.dk

**Data Availability Statement:** Data cannot be shared publicly because of confidentiality agreements between the parties involved in the study. Data are available from P.W.

## Abstract

In this study, we show that virtual reality (VR) behaviometrics can be used for the assessment of compliance and physical laboratory skills. Drawing on approaches from machine learning and classical statistics, significant behavioral predictors were deduced from a logistic regression model that classified students and biopharma company employees as experts or novices on pH meter handling with 77% accuracy. Specifically, the game score and number of interactions in VR tasks requiring practical skills were found to be performance predictors. The study provides biopharma companies and academic institutions the possibility of assessing performance using an automatic, reliable, and simple alternative to traditional in-person assessment methods. Integrating the assessment into the training tool renders such laborious post-training assessments unnecessary.

## Introduction

Employees need to be retrained at regular intervals. This is particularly crucial in industries that are highly regulated and where human error can have costly or life-threatening consequences, for example in biopharma manufacturing [1,2]. However, whether current assessment methods reflect real learning outcomes is a major debate in professional training [3]. Traditionally, employees in biopharma manufacturing are assessed post training by a theoretical compliance test. While these tests are widely accepted in the industry, experts have criticized them for measuring only knowledge retention and comprehension instead of on-the-job skills [3–5].

Concerned about the effectiveness of conventional types of assessment, the US Food and Drug Administration announced they will "shift their inspection focus to performance and away from compliance." In practice, this means that employees will have to pass a performance demonstration in which a qualified trainer assesses their on-the-job skills [4]. However, considering the current frequency of retraining and assessment, conducting such performance demonstrations would be resource intensive and expensive. We thus hypothesized that performance demonstrations could be outsourced to virtual reality (VR) as a standardized,

(wismerp@gmx.ch) or from the data access committee (pselivanov@labster.com) for researchers who meet the criteria for access to confidential data.

**Funding:** P.W. and M.O.A.S received funding from Innovation Fund Denmark (Innovationsfonden) under large-scale project, 5150-00033, SIPROS (https://innovationsfonden.dk/en). M.O.A.S received funding from the Novo Nordisk Foundation (Novo Nordisk Fonden) under NFF grant number NNF10CC1016517 (https://novonordiskfonden.dk/en/). The funders had no role in study design, data collection and analysis, decision to publish, or preparation of the manuscript.

**Competing interests:** I have read the journal's policy and the authors of this manuscript have the following competing interests: P.W. holds shares in Labster Aps that co-developed the VR simulation. S.A.S, K.A.E, and M.O.A.S have no competing interests.

inexpensive alternative to in-person assessments. A similar approach was previously taken in the medical field, where measures of errors, time, and economy of movements in the VR environment were found to be correlated to surgical expertise [6].

Indirectly predicting performance has the advantage of not biasing trainees to the assessment. Several studies have shown that predictability of the assessment can lead to surface learning [7]. For example, when trainees were given questions that were meant to induce them to take an in-depth analytic approach about text they read (e.g., What is the relationship between various subsections?), they counterintuitively showed shallower learning than those that were not given any reflective questions [8]. Hence, non-intrusive, "stealth" assessment methods using behavioral patterns are desirable from a learning standpoint [9].

The COVID-19 pandemic imposed extra burdens on professional training and the educational system in general. In April 2020, 89.4% of students worldwide were affected by school and university closures. Although more than 90% of universities from 107 countries switched to distance learning and teaching, successful tools for remote assessment are difficult to apply [10]. Hence, challenges such as academic dishonesty or the evaluation of practical skills in remote setups could be overcome by behavior-based assessments in digital environments [11].

Many research groups report that behavioral patterns observable from the use of a mouse or keyboard can vary from individual to individual and with mood or level of attention. Variation can be so pronounced that mouse usage and keystroke dynamics can be used for authentication and identification [12,13], gender recognition [14], and measuring emotions, stress and engagement in tutoring contexts [15–18].

Previously, behavioral data was used to predict students' performances in programming contexts and on a math test [19–21]. In the latter case, a model that used time spent on math problems, selection of correct or incorrect answers during game play, or other behaviors predicted students' post-test scores. In this study, we extend this approach to VR training for biopharma manufacturing that is demonstrated to be more effective than reading standard operating procedures and may be able to replace real-life training [22]. We investigated the feasibility of using behaviometrics recorded in a virtual laboratory simulation on the topic of pH calibration as a replacement for a compliance test and alternative to real-life assessment.

## Methods

### Participants

Participants were 55 pharmaceutical company employees (male: 37, female: 18; all age intervals above 20 years) of different expertise levels (industrial operators, equipment-responsible personnel, and others such as general managers) and 24 first-year students from two biopharma production schools (male: 20, female: 4; all age intervals from 10 to 50 years) who were enrolled in a tertiary education program to become industrial operators. Study participants were recruited by a pharmaceutical company from their metrology departments and associated educational institutions. 78% of participants reported that they had never tried VR before participating in this study, while 22% had used it occasionally.

The study was approved by Labster Aps that co-developed the VR simulation, and Labsters pharmaceutical company collaborator. It was carried out according to Labsters terms, conditions, and privacy policy [23]. Participants provided informed consent to the use of their personal data for research purposes. The study is exempted from IRB approval according to 45 CFR 46 set forth by the Office for Human Research Protections at the U.S. Department of Health and Human Services (HHS) [24].

## Procedure

Participants were exposed to immersive VR on a Lenovo Mirage Daydream headset. The device followed the head rotation of the player for a 360° view. The corresponding game controller was used to interact with elements of the virtual lab and navigate to different points in the environment. Participants received instructions on how to use the device and were advised to sit down during the intervention to prevent accidents.

The VR simulation was a one-hour educational game on how to operate a pH meter according to standard procedures in pharmaceutical manufacturing [22]. After completing the integrated pre-test, participants performed 146 tasks in the VR simulation, for example flushing the pH meter electrode with water from a wash bottle (Fig 1A). The tasks were interspersed with 17 challenges, distributed throughout the simulation. The challenges consisted of dialogs related to the task the participant was performing. During a challenge, participants were

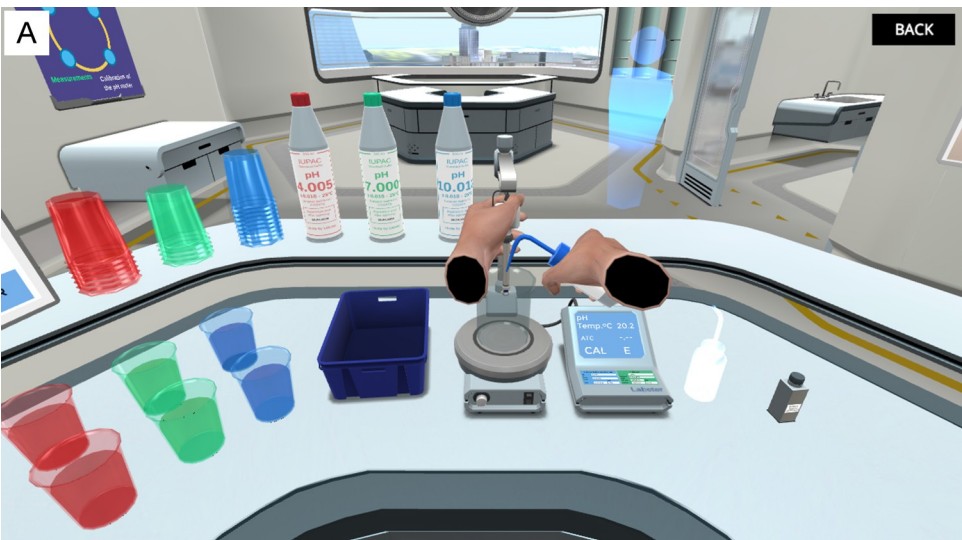

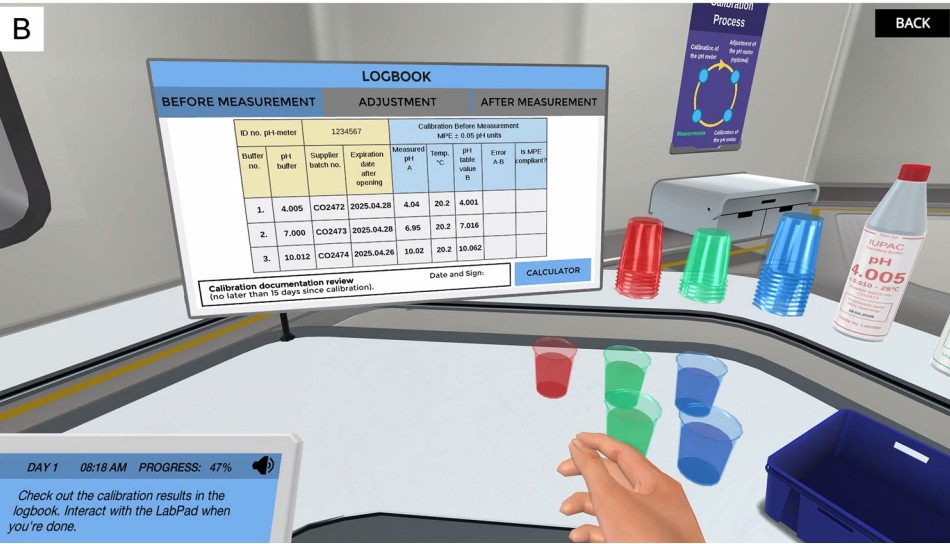

**Fig 1. Virtual reality (VR) simulation on pH meter operation.** A: VR simulation task to flush the pH meter electrode with water from a wash bottle. B: In-game challenge to evaluate pH calibration points and decide on the next steps.

presented with four options, only one of which would lead them to the next step in the simulation. For example, when faced with an erroneous reading for a pH calibration point, players had to correctly decide to adjust the pH meter to continue (Fig 1B). Throughout the simulation, participants were able to access relevant theory and instructional information on a virtual tablet.

After playing the VR simulation, the first-year students completed a theoretical compliance test and performed a physical lab demonstration. In the demonstration, practical laboratory skills were assessed by metrology experts while students individually performed the procedure from the VR simulation with real lab equipment.

## Metrics

From the VR simulation logs recorded during gameplay, a total of 340 behavioral patterns (behaviometrics) were extracted. For each of the 146 tasks, the following events were logged: time stamp, number of interactions with elements of the virtual lab (e.g., objects such as the pH meter), number of theory page views, and game score for challenge tasks. Tasks with no events for any participants (e.g., automated animations) and incomplete data records (e.g., if participants dropped out due to cybersickness) were excluded from the analysis. The behaviometrics were categorized and summarized into eight interpretable predictors (Table 1).

The lab performance test was 21 checklist items that reflected the steps in the VR simulation. The experts scored whether participants correctly performed each step. The compliance

**Table 1. Behavioral patterns from the VR simulation and self-reported personal information to predict compliance, physical lab performance and expertise.**

| Pre-test metrics | Description |
|---|---|
| Expertise | Prior knowledge, self-perceived prior knowledge, amount of training and current occupation combined. |
| Age | Age of the participant (6 categories, 10-year intervals from 10 to >60 years old). |
| Gender | Female or male. |
| VR experience | Self-reported VR experience prior to participating in this study (5 levels: from "I have never tried it before" to "I use it daily"). |
| | |
| **Behaviometrics** | **Description** |
| Practical skill interactions | Number of interactions with elements of the virtual lab in tasks requiring practical laboratory skills. |
| Practical skill time | Time spent in tasks requiring practical laboratory skills. |
| Challenge score | Score obtained in in-game challenges. |
| Challenge time | Time spent in in-game challenges. |
| Theory lookups | Number of times participants accessed the theory pages. |
| Reading interactions | Number of interactions with elements in the virtual lab while reading text. |
| Reading time | Time spent reading text. |
| Interaction speed | Number of interactions per second. |
| | |
| **Post-test metrics** | **Description** |
| Lab performance | Correctly executed tasks in the performance demonstration. |
| Compliance | Correctly answered questions in the theoretical compliance test. |

All metrics apart from age and gender were continuous and normalized. Pre-test metrics were recorded from an in-game questionnaire, behaviometrics were deduced from user logs, and post-test metrics were obtained from an online questionnaire and the performance demonstration.

test consisted of 15 multiple choice knowledge questions, each with four answer possibilities and one correct answer.

To label participants as experts or novices in pH meter operation, a preliminary question-naire (pre-test) was administered from which prior knowledge, self-perceived prior knowledge, amount of training and current occupation were combined into an average expertise score (S1 Table). All four variables of the pre-test had equal weight in the expertise score. The pre-test also recorded participants' protected attributes age and gender, as well as their prior experience in using VR.

## Statistical modeling

In this study, methodologies from both classical statistics and machine learning were used: backwards selection and analysis of covariance (ANCOVA) from classical statistics and regularization, confusion matrices and cross-validation from machine learning.

Univariate linear regression models were employed to correlate specific behaviometrics to real lab performance and compliance. This approach was chosen over more complex modeling approaches, for example multiple linear regression, due to the small sample size and cross-correlations between behaviometrics.

Two models were created to classify participants into expertise levels: a reduced logistic regression model based on the eight summarized behaviometrics (Table 1), and a regularized logistic regression model based on all available metrics (performance model). More complex machine learning approaches such as boosted trees and random forests were tested but they did not improve model performance.

Due to class imbalances, oversampling of expertise groups was applied to increase the overall model performance and improve the predictive power for the minority class (S2 Table). A combination of backwards selection and ANCOVA was used to manually select independent metrics for the reduced logistic regression model (see Results). Elastic-net regularization parameters for the performance model were deduced from a 3-fold, 10x repeated (nested) cross-validation.

All analyses were performed in the R software environment.

## Results

### Correlating in-game behaviors to compliance and physical lab performance

In this study, we collected behavioral data during an educational VR game on pH meter operation in biopharma manufacturing. All study participants self-reported their expertise on the subject in a pre-test, while a subset of participants additionally performed a physical lab demonstration and took a compliance test after completing the laboratory simulation.

We correlated summarized behavioral data (behaviometrics) to physical lab skills and compliance to discover in-game metrics that are indicative of real-life performance. Using the full dataset, we then built a prediction model to classify experts and novices in pH meter operation based on the behaviometrics.

To discover in-game behaviors that are indicative of real-life performance, we investigated which of the recorded variables from the VR simulation logs predicted compliance and physical laboratory skills. For this purpose, we conducted a physical lab performance demonstration and compliance test with a subset of participants (first-year biopharma production students) after they completed the VR simulation. We compared the results to the interpretable, summarized behaviometrics (Table 1). The analysis showed that in-game challenge score correlated with compliance ($P<0.01$) but not physical lab performance, while the number of interactions during practical, hands-on VR tasks (practical skill interactions) correlated with both

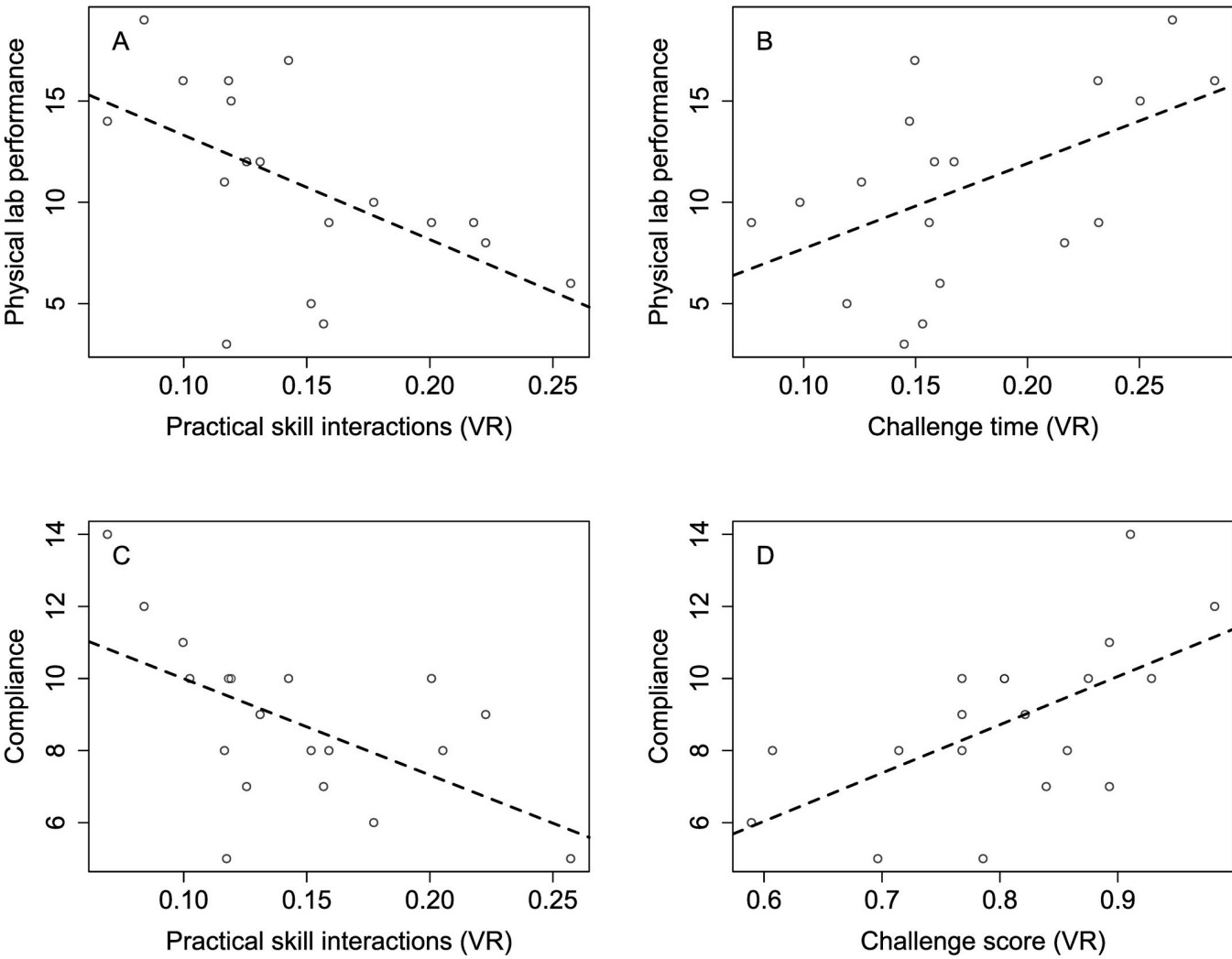

**Fig 2. Univariate linear regression models correlating physical lab performance and compliance to virtual reality (VR) simulation behaviometrics.** All statistically significant results are presented ($P<0.05$). A: Fewer interactions in the simulation in tasks requiring practical skills led to better lab performance. B: More time in simulation challenges led to better lab performance. C: Fewer interactions in the simulation in tasks requiring practical skills led to higher compliance scores. D: Higher challenge scores in the simulation led to higher compliance scores.

compliance ($P = 0.01$) and physical lab performance ($P = 0.02$). In addition, the time that participants spent in challenges (challenge time) correlated with physical lab performance ($P = 0.03$). The higher the challenge score and the lower the number of practical skill interactions, the higher the participants' compliance test result. Participants' physical lab performance increased with lower numbers of practical skill interactions and more time spent in challenges (Fig 2).

## Classifying study participants by expertise

We hypothesized that the behaviors that correlated with compliance and physical lab performance could be used to classify study participants into experts and novices in pH meter handling. We investigated if, using the full data set, we could create a more powerful prediction model beyond univariate correlations. To classify study participants by expertise, they were first binned into expertise levels according to their self-reported pre-test scores: Based on the

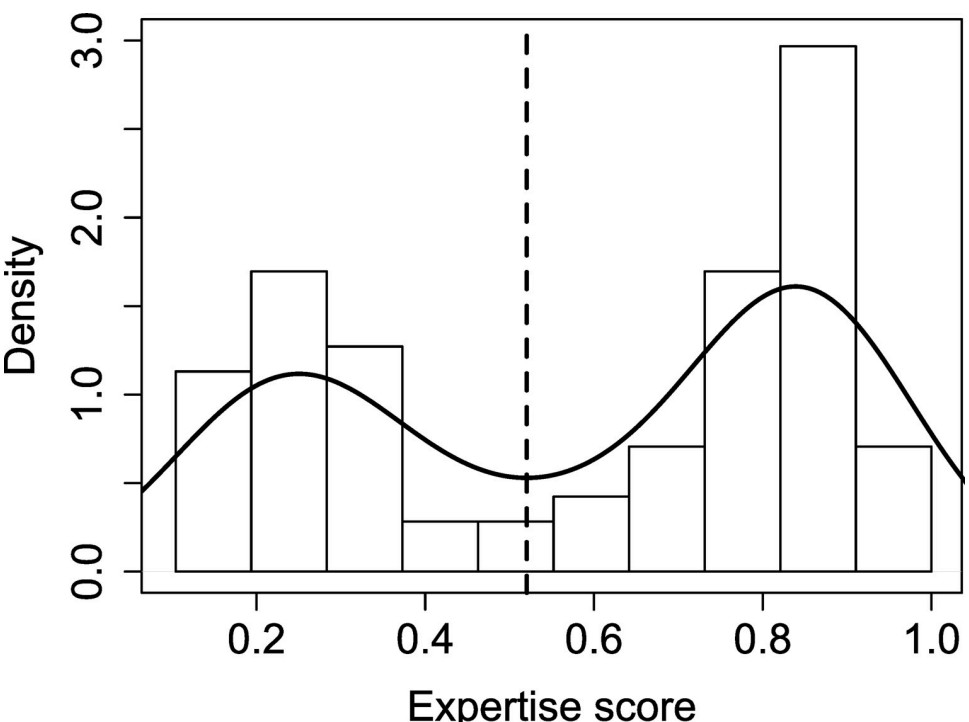

**Fig 3. Density distribution of pre-test expertise scores.** The expertise metric combined answers to questions about prior knowledge, self-perceived prior knowledge, amount of training and current occupation into an average score (S1 Table). Participants were divided into experts and novices based on the threshold at the local minimum of the distribution.

density distribution of participants, the dataset was split into two distinct expertise groups according to a threshold set at the local minimum of the curve (Fig 3). Participants with less than 0.52 points on the pre-test expertise score were considered novices and those with higher points were considered experts.

We then created an independent logistic regression model based on the summarized behaviometrics that classified study participants into the two predetermined expertise groups. Through backwards selection, we found that challenge score and practical skill interactions were highly predictive of the expertise outcome (Table 2)—the same metrics previously found to correlate with compliance and physical lab performance. Thus, the independently selected parameters of this reduced classification model are best explained by participants' practical skills and knowledge of compliance in relation to their expertise. On average, expert participants had higher challenge scores and lower numbers of practical skill interactions, which was associated with better compliance and physical lab performance.

Unbiased behavioral predictors must not be influenced by the protected attributes gender or age. When added as a covariate in the reduced classification model, gender did not have a significant influence on the prediction ($P = 0.88$). However, due to cross-correlations among

**Table 2. Behaviometric predictors of expertise groups after variable reduction.** Challenge score and practical skill interactions were significant predictors in the reduced logistic regression model.

| Behaviometric | Coefficient | Z-Statistic | P-value |
|---|---|---|---|
| Challenge score | 1.67 | 3.45 | <0.001 |
| Practical skill interactions | -3.47 | -2.86 | <0.01 |

**Table 3. Confusion matrices showing average cell counts across independent test sets for the reduced and performance logistic regression models.**

| Reduced Model | | Actual: | |
| --- | --- | --- | --- |
| | | Novice | Expert |
| Predicted: | Novice | 29% (7.6) | 13% (3.4) |
| | Expert | 10% (2.7) | 48% (12.6) |

| Performance Model | | Actual: | |
| --- | --- | --- | --- |
| | | Novice | Expert |
| Predicted: | Novice | 27% (7.1) | 7% (1.8) |
| | Expert | 12% (3.2) | 54% (14.2) |

age, the selected behaviometrics, and the expertise group, we further investigated if differences in behavior could be explained by the expertise group or age. Calculating 2 x 2 ANCOVAS (type 2) for each behaviometric separately, we exclusively found significant main effects for the expertise group but not for age, with no significant interactions between age and expertise group (S3 Table). Hence, we concluded that both main predictors used in the reduced classification model were explained by expertise alone, legitimizing their use for performance prediction.

## Model performance and evaluation

To evaluate how the reduced classification model of novices and experts generalized to unseen data, we subjected the model to a 3-fold, 10x repeated cross-validation. We also compared it to a regularized logistic regression model (performance model), built from all available metrics (340 behaviometrics, age and gender). The models' prediction accuracies across all test sets were 77% (area under the curve [AUC] = 0.80) for the reduced model, and 81% (AUC = 0.88) for the performance model as calculated from confusion matrices (Table 3). With the reduced model, an average of 74% of novices and 79% of experts were correctly classified in each independent test set. Assuming that the difference in accuracies is normally distributed, we calculated the credibility bounds of the 95% prediction interval to be -0.13 and 0.22. Hence, the classification rate of the reduced model was not significantly different from the classification rate of the performance model.

## Discussion

Our results showed that behaviometrics from VR tasks correctly predicted expert or novice status and scores correlated with compliance and performance in a real-world test of the task. Our study was based on a VR simulation of pH meter use, with behavioral data from reading, challenge and interaction tasks, and other metrics within the VR environment.

In our study, expertise groups were defined from a threshold of pre-test scores. Participants with different expertise backgrounds formed two visually distinct groups in the density distribution of scores, which was used to establish the threshold. This is in contrast to previous studies, where expertise was more normally distributed, making it difficult to differentiate distinct groups [19,25].

Our reduced statistical model based on only two predictors was able to classify novices and experts into their respective expertise groups with 77% accuracy based on behaviometrics data. The classification rate of the reduced model was not significantly lower than that of the full performance model with 342 predictors that employed machine learning approaches for model building (81% accuracy). The accuracy range is comparable to previous studies that

predicted performance from students' interaction patterns in programming courses collected over several weeks (from 70–89%) [20,26–29]. In addition, the predictors used in the reduced model are interpretable, thus adhering to the explainable artificial intelligence paradigm. They are also independent of the protected attributes of gender and age, so the applied methodology was considered fair [30].

The summarized behaviometrics (Table 1) were individually evaluated for their correlation to physical laboratory skills and compliance. In line with the reduced expertise model, the number of interactions with elements of the virtual lab in tasks requiring practical laboratory skills (practical skill interactions) negatively correlated with physical lab performance and compliance. This result can be explained by participants' trial-and-error behavior, a commonly used metric in automatic assessment environments that is correlated with worse performance [25,31]. Trainees who executed the laboratory tasks according to protocol made fewer errors and thus needed fewer steps to complete the tasks in both the virtual and physical laboratories.

Also in line with the reduced expertise model, scores for in-game challenges (challenge scores) positively correlated with compliance, indicating that the ability to solve concrete problems in the virtual laboratory was reflective of the test outcomes.

Additionally, we found that time spent on in-game challenges (challenge time) positively correlated with physical lab performance. Similar results were previously reported for engagement prediction: the more time trainees spent on executing tasks, the higher their engagement [17]. In virtual laboratories, higher engagement was shown to lead to better performance [22]. In the context of biopharma manufacturing, where accurate execution of predefined processes is critical, this result may also be explained by more thorough participants making fewer mistakes in the subsequent performance demonstration.

In conclusion, the presented approach represents a step towards implementing behaviometrics in biopharma manufacturing with a focus on replacing existing performance demonstrations and compliance tests. This approach promises a more efficient characterization of trainees' relevant skills, while reducing the cost and time spent on laborious assessments. The presented approach can also be applied for remote assessment–an add-on to remote training that is becoming increasingly popular due to the novel coronavirus pandemic at the same time that companies are starting to realize its cost and convenience benefits. Our behavior-based approach to performance assessment also solves the issue of academic dishonesty in distant training contexts [11–13].

The problem we set out to solve with this study, laborious performance demonstrations, put the same constraints on the study setup: while the pre-test was easy to administer, the collection of physical performance data was limited to a subset of participants. The presented approach could therefore serve as a guideline for similar studies, but with a greater focus on the performance demonstrations. To evaluate employees in a real-life setting, the prediction accuracies found in this study might not be sufficiently high. In desktop applications, researchers were able to differentiate individuals with an accuracy of 98% by tracking their mouse movements [12]. Hence, collecting the corresponding behavioral data in VR, for example tracking head and eye movements, might lead to similarly accurate results. This would likely allow implementing VR behaviometrics as a testing strategy on a larger scale, for example on department or company level, to investigate its long-term economic and organizational benefits.

## Supporting information

**S1 Table. Pre-test questionnaire.** Expertise metrics were used to calculate the participants' expertise scores and divide them into novices and experts.
(PDF)

**S2 Table. Overview of model parameters for the performance and reduced logistic regression models.** Model parameters were calculated for different sampling strategies. (PDF)

**S3 Table. Summary table of 2 x 2 ANCOVA comparing the influence of age and expertise group on the behavioral predictors used in the reduced logistic regression model.** No significant interactions were observed between age and expertise group. No significant main effects were observed for age. However, significant main effects were observed for the expertise group for both behaviometrics. (PDF)

## Acknowledgments

We would like to thank Line Katrine Harder Clemmensen for establishing the contact between the two research groups and for her advice. Our thanks also go to Chris Tachibana for copyediting.

## Author Contributions

**Conceptualization:** Philip Wismer.

**Data curation:** Philip Wismer, Kasper Alnor Einarson.

**Formal analysis:** Philip Wismer.

**Funding acquisition:** Morten Otto Alexander Sommer.

**Investigation:** Philip Wismer.

**Methodology:** Philip Wismer.

**Project administration:** Philip Wismer, Sarah Aparecida Soares.

**Resources:** Philip Wismer.

**Software:** Philip Wismer, Kasper Alnor Einarson.

**Supervision:** Morten Otto Alexander Sommer.

**Validation:** Philip Wismer, Sarah Aparecida Soares, Kasper Alnor Einarson.

**Visualization:** Philip Wismer.

**Writing – original draft:** Philip Wismer, Sarah Aparecida Soares, Kasper Alnor Einarson.

**Writing – review & editing:** Philip Wismer, Sarah Aparecida Soares, Kasper Alnor Einarson, Morten Otto Alexander Sommer.

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
