## [Editor Report · Decision Letter 0]

2 Jun 2022

PONE-D-22-14033

Laboratory performance prediction using virtual reality behaviometrics

PLOS ONE

Dear Dr. Sommer,

Thank you for submitting your manuscript to PLOS ONE. After careful consideration, we have decided that your manuscript does not meet our criteria for publication and must therefore be rejected.

I am sorry that we cannot be more positive on this occasion, but hope that you appreciate the reasons for this decision.

Kind regards,

Heng Luo, Ph.D.

Academic Editor

PLOS ONE

Additional Editor Comments:

It has been marked as a duplicate submission by Editorial Manager system of Plos One. 
---

## [Author Response · Author response to Decision Letter 0]

23 Jun 2022

Dear Dr. Heng Luo and Editorial Board of PLOS ONE, 

Thank you for reviewing our manuscript entitled "Laboratory performance prediction using virtual reality behaviometrics". We believe that the manuscript was marked as duplicate submission by the Editorial Manager system because it was previously included in Dr. Wismer's PhD thesis (https://backend.orbit.dtu.dk/ws/portalfiles/portal/258081313/PhDThesis_PhilipWismer_3_.pdf). Under Danish law, the thesis needs to be publicly available from the University’s online library. Hence, we would like to highlight that the presented work has not been previously published in any academic journal. Given these circumstances, we would therefore appreciate if you were willing to re-evaluate our submission.

Sincerely,

Morten Sommer & Philip Wismer

---

## [Decision Letter · Decision Letter 1]

24 Aug 2022

PONE-D-22-14033R1Laboratory performance prediction using virtual reality behaviometricsPLOS ONE

Dear Dr. Sommer,

Thank you for submitting your manuscript to PLOS ONE. After careful consideration, we feel that it has merit but does not fully meet PLOS ONE’s publication criteria as it currently stands. Therefore, we invite you to submit a revised version of the manuscript that addresses the points raised during the review process.

We look forward to receiving your revised manuscript.

Kind regards,

Walid Kamal Abdelbasset, Ph.D.

Academic Editor

PLOS ONE

Journal Requirements:

Additional Editor Comments (if provided):

Reviewers' comments:

Reviewer's Responses to Questions

**Comments to the Author**

1. If the authors have adequately addressed your comments raised in a previous round of review and you feel that this manuscript is now acceptable for publication, you may indicate that here to bypass the “Comments to the Author” section, enter your conflict of interest statement in the “Confidential to Editor” section, and submit your "Accept" recommendation.

Reviewer #1: (No Response)

2. Is the manuscript technically sound, and do the data support the conclusions?

Reviewer #1: Yes

3. Has the statistical analysis been performed appropriately and rigorously? 

Reviewer #1: Yes

4. Have the authors made all data underlying the findings in their manuscript fully available?

Reviewer #1: Yes

5. Is the manuscript presented in an intelligible fashion and written in standard English?

Reviewer #1: Yes

6. Review Comments to the Author

Reviewer #1: The present study evaluates the feasibility of using VR behaviometrics in biopharma manufacturing as a replacement for a compliance test and alternative to real-life assessment. The experimental studies are overall well designed and the data analysis appears to be robust. The experiments show that this approach represents a step towards replacing existing performance demonstrations and compliance tests. To make it more understandable, some aspects of the paper can be improved:

- In the “Procedure” section add an overview of the VR simulator of pH meter use.

- Add some images for the tasks performed in VR simulation;

- Specify the participants’ age range and previous experience of using VR simulators.

- Indicate whether the VR simulator caused participants to experience any negative effects related to cybersickness.

- Provide some details about future work;

Regards

7. PLOS authors have the option to publish the peer review history of their article (what does this mean?). If published, this will include your full peer review and any attached files.

Reviewer #1: No

---

## [Author Response · Author response to Decision Letter 1]

22 Sep 2022

Reviewer #1

We thank the reviewer for their excellent comments, which we have addressed as detailed below to improve the manuscript:

“In the “Procedure” section add an overview of the VR simulator of pH meter use.”

We have added a short paragraph on the VR simulator that was used at the beginning of the “Procedure” section (lines 81-85).

“Add some images for the tasks performed in VR simulation”

We have included a new figure (Fig 1, line 100) showcasing the two tasks mentioned in the manuscript: flushing the pH electrode with water and evaluating different pH calibration points. We have added references to the figure on lines 89 and 93. The numbering of existing figures was adjusted accordingly.

“Specify the participants’ age range and previous experience of using VR simulators”

We have added the age range of participants on line 68: “pharmaceutical company employees (male: 37, female: 18; all age intervals above 20 years)” and line 71: “first-year students from two biopharma production schools (male: 20, female: 4; all age intervals from 10 to 50 years)”.

We have added the self-reported experience with VR as part of the metrics table (Table 1, line 119) and included the following reference on line 73: “78% of participants reported that they had never tried VR before participating in this study, while 22% had used occasionally.”

“Indicate whether the VR simulator caused participants to experience any negative effects related to cybersickness.”

The VR simulation was designed to reduce cybersickness. However, there were a couple of participants that could not continue the VR simulation for this reason. These participants were disregarded for the analysis. Line 108: “incomplete data records (e.g., if participants dropped out due to cybersickness) were excluded from the analysis.”

“Provide some details about future work”

We added a new paragraph at the end of the discussion section to address limitations and further directions (lines 262-271).

---

## [Decision Letter · Decision Letter 2]

5 Dec 2022

Laboratory performance prediction using virtual reality behaviometrics

PONE-D-22-14033R2

Dear Dr. Sommer,

We’re pleased to inform you that your manuscript has been judged scientifically suitable for publication and will be formally accepted for publication once it meets all outstanding technical requirements.

Kind regards,

Walid Kamal Abdelbasset, Ph.D.

Academic Editor

PLOS ONE

Additional Editor Comments (optional):

Reviewers' comments:

Reviewer's Responses to Questions

**Comments to the Author**

1. If the authors have adequately addressed your comments raised in a previous round of review and you feel that this manuscript is now acceptable for publication, you may indicate that here to bypass the “Comments to the Author” section, enter your conflict of interest statement in the “Confidential to Editor” section, and submit your "Accept" recommendation.

Reviewer #1: All comments have been addressed

Reviewer #2: All comments have been addressed

2. Is the manuscript technically sound, and do the data support the conclusions?

Reviewer #1: (No Response)

Reviewer #2: Yes

3. Has the statistical analysis been performed appropriately and rigorously? 

Reviewer #1: (No Response)

Reviewer #2: Yes

4. Have the authors made all data underlying the findings in their manuscript fully available?

Reviewer #1: (No Response)

Reviewer #2: Yes

5. Is the manuscript presented in an intelligible fashion and written in standard English?

Reviewer #1: (No Response)

Reviewer #2: Yes

6. Review Comments to the Author

Reviewer #1: (No Response)

Reviewer #2: Dear authors,

I really appreciate all your efforts to address all the comments in a very positive manner.

Now the article is good enough to publish in the current state.

Regards

7. PLOS authors have the option to publish the peer review history of their article (what does this mean?). If published, this will include your full peer review and any attached files.

Reviewer #1: No

Reviewer #2: **Yes: **Dr. Gopal Nambi, Prince Sattam bin Abdulaziz University, Al Kharj, Saudi Arabia

---

## [Editor Report · Acceptance letter]

8 Dec 2022

PONE-D-22-14033R2 

Laboratory performance prediction using virtual reality behaviometrics 

Dear Dr. Sommer:

I'm pleased to inform you that your manuscript has been deemed suitable for publication in PLOS ONE. Congratulations! Your manuscript is now with our production department. 

Kind regards, 

on behalf of

Dr. Walid Kamal Abdelbasset 

Academic Editor

PLOS ONE